# Unusual Canine Cutaneous Melanoma Presenting Parietal Bone Metastasis: A Case Report

**DOI:** 10.3390/vetsci10040282

**Published:** 2023-04-06

**Authors:** Ozana-Maria Hritcu, Florentina Bocaneti Daraban, Fabian Dominic Bacusca, Aurelian-Sorin Pasca

**Affiliations:** Faculty of Veterinary Medicine, Iasi University of Life Sciences Ion Ionescu de la Brad, Mihail Sadoveanu Alley, No.8, 700489 Iasi, Romania; florentinabocaneti@uaiasi.ro (F.B.D.);

**Keywords:** melanoma, cutaneous, VEGF, MMP-2, MMP-10

## Abstract

**Simple Summary:**

This report aims to investigate a case of canine cutaneous melanoma with generalized metastasis and an unusual parietal bone metastasis, never reported before to the authors’ knowledge. The primary tumour was located on the skin of the carpal region and tumoral cells migrated towards the regional lymph node, reaching the general circulation through the blood and lymphatic streams. The histopathological examination showed a mixture of spindle, epithelioid and dendritic melanocytes, with variations in terms of pigmentation, a mitotic count of 10 (calculated for 10 high-power fields) for the primary tumour and higher for the metastasis. Immunohistochemistry was used to screen the expression of matrix metalloproteinases −2, −10 and the vascular endothelial growth factor (VEGF), all considered important invasiveness factors that facilitate the modelling of the extracellular matrix to enable invasiveness and metastasis. The results showed a strong positive reaction for VEGF and MMP-10, and a moderate positivity for MMP-2. This report suggests that metastasis in canine cutaneous melanomas may be present in various locations, such as the parietal bone, and is facilitated not only by the migratory abilities of melanocytes, but also by invasiveness factor expression, such as matrix metalloproteinases, MMP-2 and MMP-10, or VEGF.

**Abstract:**

Melanocytic tumour anatomic location is considered an important prognostic indicator. The cutaneous forms are generally considered benign and may show various biological behaviours. This work reports a rare case of canine cutaneous melanoma showing parietal bone metastasis. Bone invasion in melanocytic tumours is often described in oral or visceral melanomas, but not in cutaneous forms. The patient (dog, male, mixed breed, 12 years) was initially presented for the surgical removal of a cutaneous tumour located on the skin of the carpal region of the right forelimb. Four months after, the patient returned with enlarged lymph nodes and acute respiratory failure. The patient was euthanized due to a decline in physical condition. The necropsy showed metastases in the affected forelimb, regional lymph node, splanchnic organs, parietal bone and meninges. Histopathological examination of tumour tissue samples revealed a mixture of pigmented and non-pigmented spindle and epithelioid melanocytes, while according to immunohistochemistry, the tumours showed a strong immunopositivity for VEGF and MMP-10, and a moderate positivity for MMP-2 expression. This case shows that cutaneous melanocytic tumours may show an aggressive malignant form with positive immunohistochemical reactions for multiple invasiveness factors.

## 1. Introduction

Melanocytic tumours originate primarily from melanocytes, cells dispersed within the basal layer of the epidermis, dermis, mucosae and eye structures [1,2].

In animals, the incidence of melanocytic tumours is higher for melanomas (76.9%) compared to melanocytomas (23.1%), while in dogs the occurrence represents 7% of all malignant neoplasms [2]. Different studies report that mixed breeds with black hair, with ages ranging from 8 to 11 years, are the most affected, but there also seems to be an increased incidence in purebred dogs such as Schnauzer, Chow-Chow, Sharpei, Golden Retriever, Labrador Retriever, Cocker Spaniel and some Terriers [3,4,5]. Moreover, males are more affected than females [1,2]. Histologically, melanomas can be classified into six histotypes depending on the cytomorphological features: epithelioid (round and polygonal cells), spindle cell (tumour resembles fibroblasts), signet ring, balloon, dendritic and mixt tumours (showing more than one cell type). Depending on the presence of pigmentation, two types are distinguished: melanotic and amelanotic [1].

Cutaneous melanocytic tumours are generally considered benign neoplasms that can be treated with surgical excision. However, highly malignant forms have been described which in turn may be explained by the migratory nature of melanocytes and partly by the release of factors that act on the extracellular matrix degradation, allowing for invasiveness and metastasis [6,7]. Melanomas have been shown to form metastasis in the bone (mandible, femur), especially when they have an oral localization [8,9].

Besides the migratory features of melanocytes, the invasiveness of melanomas can be facilitated by other factors as well, such as enzymes that change the permeability of the extracellular matrix to allow for easy cellular infiltration. Matrix metalloproteinases are enzymes specialized in degrading the extracellular matrix components and basal membranes, facilitating the advance of the tumoral cells through the healthy tissue. They are classified into several types: collagenases (MMP-1, MMP-8, MMP-13, MMP-18), stromelysins (MMP-3, MMP-10, MMP-11), matrilysins (MMP-7, MMP-26), metallo-elastase (MMP-15, MMP-16, MMP-17, MMP-24, MMP-25), membrane-type MMPs (MMP-14, MMP-17, MMP-24, MMP-25), gelatinases (MMP-2, MMP-9) and others (MMP-12, MMP-19, MMP-20, MMP-21, MMP-23, MMP-27, MMP-28) [10]. VEGF (vascular endothelial growth factor) is a heparin-binding glycoprotein involved both in the stimulation of endothelial cell migration and proliferation, and allowing for increased vascular permeability [11].

All these factors have been demonstrated to be expressed by different types of tumours, and it was suggested that they contribute to the reorganization of the extracellular matrix in order to facilitate invasion and metastasis. The purpose of this work was to investigate the expression of MMP-2, MMP-10 and VEGF in a case of canine cutaneous melanoma with accelerated progression. Moreover, to our knowledge, this is the first report of a canine cutaneous melanoma resulting in metastasis in the parietal bone.

## 2. Materials and Methods

The patient (dog, male, mixed breed, 12 years) was initially presented for the surgical removal of a cutaneous tumour, which, after histopathological examination, was diagnosed as a melanoma. The primary tumour was located on the skin of the carpal region of the right forelimb, presenting a nodular shape and a diameter of about 2 cm, pigmented and well-circumscribed, with a calculated mitotic count of 10. Four months later, the patient returned with enlarged lymph nodes and acute respiratory failure. Euthanasia was recommended and performed using the following protocol based on the recommendations of the World Society for the Protection of Animals and the American Veterinary Medicine Association and agreed upon by the owner [12]: atropine IM injection (0.044 mg/kg), xylazine IM injection 15 min after (1.1 mg/kg), ketamine IM injection 5 min after (22 mg/kg) and an embutramide solution slowly administered intravenously (0.5 mL/kg).

Necropsy was performed and tissue samples were collected for histopathological examination, fixed in 10% formalin solution and processed using the paraffin embedding method as follows: embedding in paraffin using a Leica TP 1020 automatic tissue processor (Leica Biosystems, Manheim, Germany), Slee MPS paraffin embedding centre and a cold plate (Slee, Olm, Germany). Sectioning was done at 3 µm using a Slee Cut 5062 microtome(Slee, Olm, Germany). The slides were stained using the Masson trichrome staining method and mounted using a clear mounting medium.

Further, the slides were evaluated using a Leica DM750 optical microscope equipped with a Leica ICC50HD camera (Leica Biosystems, Manheim, Germany). The mitotic count was calculated by analysing 10 random high-power fields = microscopic fields with magnification power ×400 (HPF).

Immunohistochemistry was performed on paraffin-embedded tissue samples using the detection kit, Novolink Max Polymer Detection system (Leica). The immunohistochemical protocol was described by authors in a previous study [13] and slightly adapted as follows: two deparaffination baths in xylene (20 min each), treatment with 97% ethylic alcohol for 10 min, denatured ethylic alcohol bath for 10 min, bleaching in a 10% H_2_O_2_ solution heated at 60 °C, for 60 min (to eliminate the melanin which would interfere with the diaminobenzidine (DAB) staining), treatment for the inactivation of peroxidase in a H_2_O_2_—methanol solution for 20 min, hydration with three successive alcohol baths at 90 °C, 80 °C and 50 °C, and distilled water for 5 min each. The proteolytic treatment was performed in a pH 6.00 citrate buffer. Following this treatment, two phosphate-buffered saline (PBS) pH 7.4 washes (5′ each) and serum blocking for 20 min were performed. The primary antibodies, anti-VEGF (1:500, sc-53462, Santa Cruz, Dallas, TX, USA), anti-MMP-10 (1:200, sc-374029, Santa Cruz, Dallas, TX, USA) and anti-MMP-2 (1:200, Invitrogen, Rockford, IL, USA) were applied overnight at 4 °C. The next day, the formalin-fixed paraffin-embedded (FFPE) sections were washed twice in PBS (5′ each), and then incubated with the secondary antibody at room temperature for 30 min. To visualize the specific immunoreactivity, the FFPE sections were incubated with diaminobenzidine (DAB) and then counterstained with haematoxylin for 5 min. Next, the slides were dehydrated (three successive baths of 50°, 80° and 90° ethylic alcohol for 2 min each, two successive baths in denatured alcohol and absolute ethylic alcohol for 5 min each), cleared (two successive baths in xylene for 15 min each) and mounted with a clear mounting media. Negative control sections were also performed, where primary antibodies were omitted and replaced with phosphate-buffered saline instead. MMP-2, -10 and VEGF immunoreactivity were scored as previously described [13]: negative, + weak, ++ moderate, +++ strong positivity. The immunoreactivity was scored by three observers (OMH, FDB, ASP), under blind conditions.

## 3. Results

Necropsy showed metastases had formed in the carpal and metacarpal regions of the affected limb, the suprascapular lymph node, which was enlarged, the internal lymph nodes, kidneys (Figure 1a), lungs (Figure 1b), pleura, liver, pancreas, spleen, heart (Figure 1c), adrenal glands, intestine and parietal bones (Figure 1d). Microscopical examination showed that the secondary tumours had pigmented and non-pigmented melanocytes, with a mixture of spindle and epithelioid cells, showing anisocytosis, anisokaryosis and multiple mitosis, with a lower mitotic count compared to the one calculated for the primary melanoma, which was 10. Specifically, the parietal bone was characterized by tumoral melanocytes infiltrating the marrow (Figure 2a), while in the intestine, the melanoma metastasis was infiltrating the intestinal wall up to the enterocytes (Figure 2b). Interestingly, the cardiac melanoma metastasis was concentrated near the capillaries (Figure 2c). A mixture of pigmented and amelanotic cells, with oval, spindle and epithelioid shapes, showing anisokaryosis and mitosis, were often observed (Figure 2d).

The immunohistochemistry showed a strong positive reaction for VEGF (Figure 3a) and MMP-10 (Figure 3c), and a moderate response for MMP-2 (Figure 3b). The negative controls, where the first antibodies were omitted, did not show any specific immunopositivity (Figure 3d).

## 4. Discussion

The evolution of the neoplastic disease in the next 4 months was rapid, although studies report that cutaneous melanocytic tumours rarely result in distal metastasis and are more likely to affect only the regional lymph nodes [4]. Canine mixt type melanoma is considered more similar to the pigmented epithelioid melanocytoma seen in human counterparts [4].

In this case, the metastasis was generalized, of different sizes and shapes and intensely pigmented from a macroscopical point of view. The suprascapular lymph node of the right side was transformed into a tumoral mass, with tumoral emboli inside the lymphatic vessels and sinuses indicating a lymphatic pathway. This is considered the most common path for the formation of metastasis, since tumoral cells that access the extracellular matrix, beneath the basal membrane, will have access to the lymphatic vessels [14].

The metastases were located on all serosa and were infiltrative, affecting the underlying organs. In one study, serosal metastases are mentioned as a frequent finding in melanotic disease [15]. In the case of intestinal metastases, it was observed that the main part of the tumour was located in the serosa, infiltrating the submucosa and mucosa, while the enterocytes showed melanin granules. This characteristic is considered to be the result of the embryonic origin of melanocytes, since they are migratory cells of a dendritic type which express genes associated with motility, allowing them to penetrate the more compact tissue types or to overpass basal membranes [14]. Moreover, intestinal submucosal metastasis is considered to be more frequent in cutaneous melanomas than in other types of cancers [14].

The cardiac metastases were located perivascularly, with infiltrative areas that followed the connective stroma of the myocardium. The same perivascular pattern of melanotic metastases was also seen in the renal parenchyma, suggesting perivascular migration. Tumoral emboli were observed both in the renal blood vessels and inside the glomeruli, accounting for the circulatory pathway. All four paths were considered (lymphatic, vascular, intracoelomic and extravascular migration) as possible mechanisms for the formation of metastases [14].

Although the pigmentation degree was similar in the primary and secondary tumours from a macroscopical point of view, histologically, it was possible to observe that the metastases contained areas with tumoral cells that were either less pigmented or amelanotic. A similar pattern was described by Smith et al. in 2002 [15].

Pulmonary metastasis was considered common, given the fact that this location is one of the most frequent sites for secondary tumours for cutaneous melanomas [14].

The most surprising metastatic site was the parietal bone and to our knowledge, this is the first report of a canine cutaneous melanoma metastasizing in this location. The melanocytes were infiltrating the meninges from within the bone. Osseous metastasis is most commonly found in oral melanomas and affects the mandible or maxillary bones. However, some studies mention this location both in humans [16,17] and dogs [8], but usually, only the meninges are affected.

The immunohistochemistry showed a strong positive reaction for VEGF, suggesting an angiogenic phenotype, which could be an indicator of rapid growth, but also easy future metastasis [11]. Interestingly, this result is in agreement with other authors’ findings, where melanomas have been reported to express a strong VEGF immunoreactivity [18]. The expression of MMP-2 has two important roles. First, it is actively correlated with the synthesis of VEGF in melanocytes [19], thus stimulating angiogenesis and the formation of vascular tumoral emboli. Secondly, it is an indicator of expansion, this gelatinase being involved in reducing cellular cohesion and stimulating cellular migration. A strong MMP-2 response is usually correlated with a low prognostic and a high tissular dysplasia, aplasia and tumoral progression [20]. However, the immunoreactivity in this case was moderate and similar to other studies [19]. For MMP-10, studies indicated that melanomas tend to express this stromelysin, especially in the radial growth phase, with a decrease in the vertical phase [21]. Similar to MMP-2, MMP-10 stimulates neoangiogenesis, especially the formation of endothelial cell tubes, creating proper conditions for vascular metastasis [22].

## 5. Conclusions

Although cutaneous melanocytic tumours are usually considered benign because of their localization, the highly malignant forms can show invasive behaviour, with the contribution of various factors that change the characteristics of the extracellular matrix, such as matrix metalloproteinases. These enzymes can allow for a rapid dispersion of tumoral cells, even inside bones that have no physical contact with the primary tumour. This process may follow any of the known metastasis pathways (circulatory, lymphatic, intracaelomic, perivascular), which means that even the sampling of sentinel lymph nodes that shows negative results may not be sufficient for a positive prognosis. Moreover, the release of factors (VEGF, MMP-2, MMP-10) that stimulate neoangiogenesis may contribute to and accelerate the formation of tumoral emboli.

## Figures and Tables

**Figure 1 vetsci-10-00282-f001:**
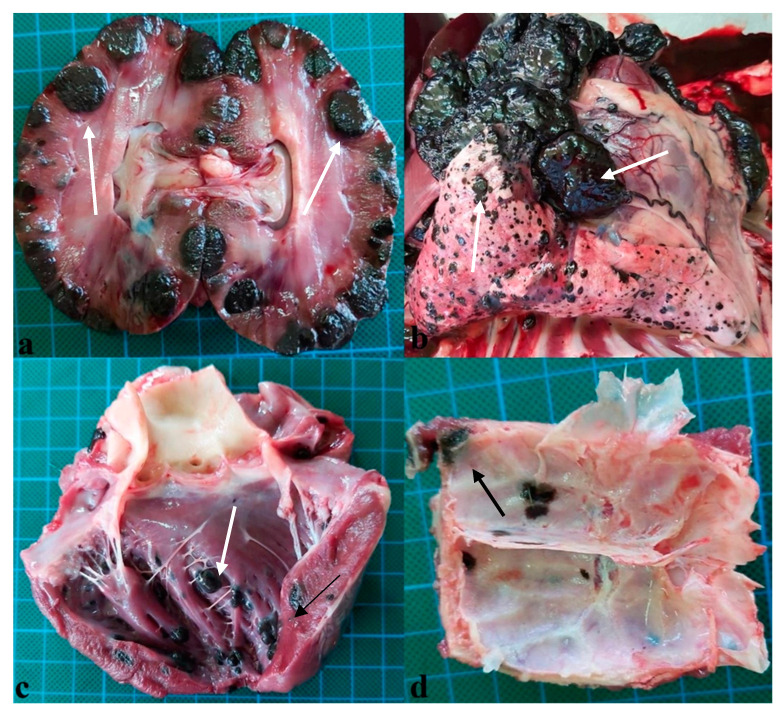
Primary melanoma in a dog. (**a**) Metastasis in kidneys (white arrows). (**b**) Pleural and lung metastasis (white arrows). (**c**) Cardiac metastasis (white arrows). (**d**) Metastasis in the parietal bone (black arrow).

**Figure 2 vetsci-10-00282-f002:**
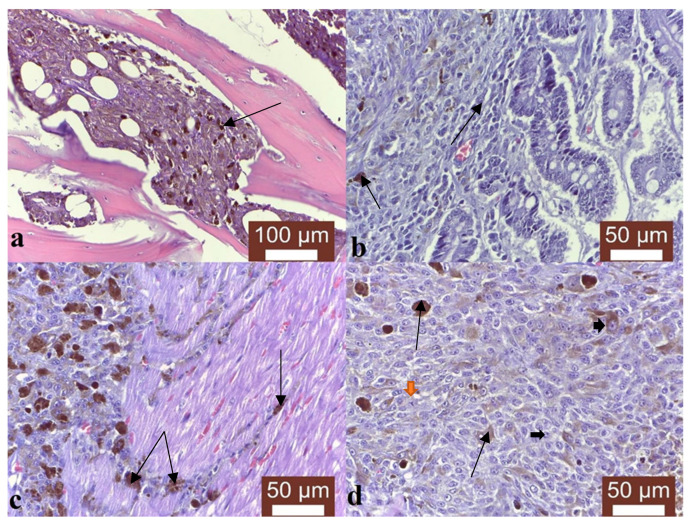
Primary melanoma in a dog. (**a**) Tumoral melanocytes infiltrating the marrow in the parietal bone (black arrow), Masson trichrome stain, ×200. (**b**) Melanoma metastasis infiltrating the intestinal wall up to the enterocytes (black arrows), Masson trichrome stain, ×400. (**c**) Cardiac melanoma metastasis concentrated near the capillaries (black arrows), Masson trichrome stain, ×400. (**d**) Mixture of pigmented and amelanotic cells (black arrows), with oval, spindle and epithelioid shapes, showing anisokaryosis (orange arrowheads) and mitosis (black arrowheads), Masson trichrome stain, ×400.

**Figure 3 vetsci-10-00282-f003:**
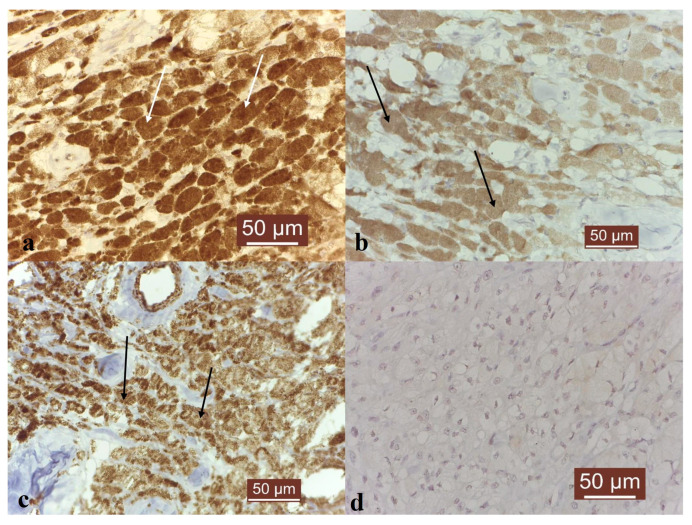
MMP-2, MMP-10 and VEGF immunoexpression in canine melanoma. Immunohistochemistry results: (**a**) Strong VEGF immunoexpression in canine melanoma (white arrows), ×400. (**b**) Moderate MMP-2 immunoexpression in canine melanoma (black arrows), ×400. (**c**) Strong MMP-10 immunoexpression in canine melanoma (black arrows), ×400. (**d**) Negative control. No specific immunopositivity, ×400.

## Data Availability

Not applicable.

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
