# Peer review of "Unusual Canine Cutaneous Melanoma Presenting Parietal Bone Metastasis: A Case Report"

_vetsci, 2023, doi:10.3390/vetsci10040282_

Round 1

Reviewer 1 Report

Firstly I would like to thank you for inviting me to review the manuscript entitled: Unusual canine cutaneous melanoma presenting parietal bone metastasis – a case report

The authors present a rare case of canine cutaneous melanoma with aggressive behavior.

The title accurately reflects the case. The case involves an important area of health and presents a clear and clinically useful message. The manuscript has a logical construction. The discussion section explains the case in the context of published information. The conclusions accurately and clearly explain the main clinical message. The figures are of good quality and relevant to the clinical message. The references are appropriate and current.

Minor comments:

1) The manuscript needs to be improved in terms of clarity, style, and use of English. I recommend professional editing.

2) In the simple summary section, the authors should add on line 14 that the mitotic index was on 10 high-power fields.

3) In lines 29 and 200, replace metastasation with metastasis.

4) In line 47, the sentence the incidence of melanomas is higher for melanomas (76.9%) compared to 47 melanocytomas (23.1%) does not make sense.

5) In lines 56 and 57, in the sentence “Depending on the intensity of pigmentation two types 56 are distinguished: melanotic and amelanotic” replace the word intensity with “presence”.

6) In line 84, replace the sentence “with visible metastases in the regional lymph node” with enlarged lymph nodes.

Author Response

Response to Referee  #1

We thank referee #1 for his/her valuable suggestions that will certainly contribute to improving the quality of our paper.

Please find below the responses to the referee, point by point.

  1. The manuscript needs to be improved in terms of clarity, style, and use of English. I recommend professional editing.

Response: We thank the referee for this observation. We re-examined the manuscript and tried to formulate the text in a more clear and correct language.

  1. In the simple summary section, the authors should add on line 14 that the mitotic index was on 10 high-power fields.

Response: Thank you for this observation, we completed the summary with this information. Please see lines 15-16.

  1. In lines 29 and 200, replace metastasation with metastasis.

Response: Thank you to the referee for this observation, we misused the word. Therefore we replaced it as suggested throughout the manuscript. Please see lines 78, 245, 257.

  1. In line 47, the sentence the incidence of melanomas is higher for melanomas (76.9%) compared to 47 melanocytomas (23.1%) does not make sense.

Response: We apologize for this error. Indeed, looking more carefully at this sentence, it does not make any sense. We reformulated it and now it is more clear. Please see line 62.

  1. In lines 56 and 57, in the sentence “Depending on the intensity of pigmentation two types 56 are distinguished: melanotic and amelanotic” replace the word intensity with “presence”.

Response: Thank you for this observation. The change was done. Please see line 72.

  1. In line 84, replace the sentence “with visible metastases in the regional lymph node” with enlarged lymph nodes.

Response: Thank you for this suggestion. We made the change accordingly. Please see lines 49, 112.

Reviewer 2 Report

The paper refers a case of metastatic cutaneous melanoma, presenting parietal bone metastasis. Although the literature on canine melanomas is very extensive, the case described is interesting both for the aggressiveness of the tumor and for the particular localization of the metastases. The cited literature appears to be up-to-date, the text is written in sufficiently correct English, only some corrections can improve it. In conclusion I believe that the manuscript can be accepted after a minor revision.

Minor changes:

Simple summary line 14: "a mitoci index" should be changed in "a mitotic count";

Abstract.

Line 24. "an important indicator" could be changed in "an important prognostic indicator";

Line 28 ""by the relased factor" could be changed in "by the relase of factors".

Introduction

Line 45. "Melanomas originated" should be chanmged in "Melanocitic tumours originated";

Line 47 " The incidence of melanomas" cshould be changed in "incidence of melanocitic tumours".

Materials and methods

Line 95. "mitotic index" should be chamnged in "mitotic count"

Lines 102 and 103: the temperatures should be corrected.

Results

Line 121 and 128. "mitotic index " should be correted in mitotic count;

Line 132. "d" after (figure 2c) must be deleted.

Discussion

Line 193. "the first report of a cutaneous melanoma" shoudl be changed in "the first report of a canine cutaneous melanoma".

Author Response

Response to Referee  #2

We thank referee #2 for his/her valuable suggestions that will certainly contribute to improving the quality of our paper.

Please find below the responses to the referee, point by point.

The paper refers a case of metastatic cutaneous melanoma, presenting parietal bone metastasis. Although the literature on canine melanomas is very extensive, the case described is interesting both for the aggressiveness of the tumor and for the particular localization of the metastases. The cited literature appears to be up-to-date, the text is written in sufficiently correct English, only some corrections can improve it. In conclusion I believe that the manuscript can be accepted after a minor revision.

Minor changes:

Simple summary line 14: "a mitoci index" should be changed in "a mitotic count";

Response: We apologize for this mistake. We have corrected it. Please see line 15.

Abstract.

Line 24. "an important indicator" could be changed in "an important prognostic indicator";

Response: We thank the referee for this suggestion which was included in the text. Please see line 43.

Line 28 ""by the relased factor" could be changed in "by the relase of factors".

Response: We thank the referee for this observation.  We have eliminated this sentence from the abstract and changed it acordingly in the introduction. Please see line 77.

Introduction

Line 45. "Melanomas originated" should be chanmged in "Melanocitic tumours originated";

Response: We thank the referee for this suggestion. We have done the change in the text. Please see line 60.

Line 47 " The incidence of melanomas" cshould be changed in "incidence of melanocitic tumours".

Response: We apologize for this misuse of this term. We have corrected it in text. Please see line 62.

Materials and methods

Line 95. "mitotic index" should be chamnged in "mitotic count"

Response: Thank you for this suggestion. We have made the correction in the text. Please see lines 111, 131.

Lines 102 and 103: the temperatures should be corrected.

Response: We thank the referee for this correction. We have addressed this issue. Please see lines 139, 142, 146, 151.

Results

Line 121 and 128. "mitotic index " should be corrected in mitotic count;

Response: Thank you for this suggestion. We have made the correction in the text. Please see line 169.

Line 132. "d" after (figure 2c) must be deleted.

Response: We apologize for this mistake. We have corrected it. Please see line 174.

Discussion

Line 193. "the first report of a cutaneous melanoma" shoudl be changed in "the first report of a canine cutaneous melanoma".

Response: We apologize for this omission. We have made the required changes. Please see line 238.

Reviewer 3 Report

Dear Authors, the article (case report) is yet another case of melanoma, but in the title you have expertly highlighted this unusual skin melanoma local invasion which deserves to be communicated to the relevant scientific community. 

The article must be improved in the chapter 2. Materials and Methods writing in depth  the embedding and sectioning histology techniques (e.g. histo section thickness), and adding IHC negative control methods,  as well as  identyfing the image capture system.

The article sound better with dedicate a chapter to the clinical examination by extracting it from chapters 2. and 3.

It will interesting understand if IHC positivity, recognized in amelanotic cells (undifferentiated cells), it is present also in melanotyc ones. A bleaching approach can be useful to clarify what melanocyte cells are involved in metastatic process (amelanotic, melanotic, or both). 

It is advisable to write a scientific article in an impersonal form. 

Details:

line 21 improve the phrase "invasiveness factors" with "invasiveness factor s expression";

line 23 improve the phrase "while the location" with "while anatomic location";

line 25 change "internal melanoma" with "visceral melanoma";

line 29 change "metastisation" with "metastasis" and provide for replacement throughout the text;

line 30 insert the indefinite article "a" between the words "report " and "rare";

line 32 identify the carpal region inserting "of the right forelimb";

line 35 change "internal" with "splanchnic";

line 53 change "categories" with "hystotype";

line 54 change "characteristic" with "features";

lines 78-79 revising the sentence "forming?"

line 91 change the sentence "A complete necropsy was performed, during which tissue sample were harvested" with the sentence "Necropsy was performed and tissue samples were collected";

line 136-139 the sentence must be moved in chapter 2. Materials and Methods. 

line 137 change "manifesting" with "showing";

line 166 change "dissemination" with "pathway";

Author Response

Response to Referee  #3

We thank referee #3 for his/her valuable suggestions that will certainly contribute to improving the quality of our paper.

Please find below the responses to the referee, point by point.

  1. The article must be improved in the chapter 2. Materials and Methods writing in depth the embedding and sectioning histology techniques (e.g. histo section thickness), and adding IHC negative control methods, as well as  identyfing the image capture system.) –

Response: We apologise for these omissions and we introduced in the sections the missing information. Please see lines 125-129, lines 153-155, line 131.

  1. The article sound better with dedicate a chapter to the clinical examination by extracting it from chapters 2. and 3.) –

Response:  We thank to the referee for his/her observation. Accordingly, we created a new paragraph in M&M section, where, the clinical examination was described, as suggested. Please see lines 109-111, 119-123.

  1. It will interesting understand if IHC positivity, recognized in amelanotic cells (undifferentiated cells), it is present also in melanotyc ones. A bleaching approach can be useful to clarify what melanocyte cells are involved in metastatic process (amelanotic, melanotic, or both).)

Response: To address the problem of the cells being pigmented in a dark brown colour which would interfere with the DAB staining used in immunohistochemistry we used a blanching protocol described in the M&M section. In the images included in the manuscript we can see that all melanocytes, both melanotic and amelanotic, manifest positivity. Please see lines 138-140.

  1. It is advisable to write a scientific article in an impersonal form.

Response: We thank to the referee for this observation. He/she is right and accordingly, we changed throughout the plain manuscript the personal form into an impersonal form. Please see the track.

Details:

  1. line 21 improve the phrase "invasiveness factors" with "invasiveness factor s expression";

Response: Thank you for the suggestion. We included it in the text. Please see line 23.

  1. line 23 improve the phrase "while the location" with "while anatomic location";

Response: We thank the referee for this correction and we added it to the text. Please see line 43.

  1. line 25 change "internal melanoma" with "visceral melanoma";

Response: We thank the referee for this observation. The correction was made at line 46.

  1. line 29 change "metastisation" with "metastasis" and provide for replacement throughout the text;

Response: We apologize for the missuse of this word. It was since corrected throughout the manuscript. Please see lines: 78, 245, 257.

  1. line 30 insert the indefinite article "a" between the words "report " and "rare";

Response: We apologize for this error. We have corrected it. Please see line 45.

  1. line 32 identify the carpal region inserting "of the right forelimb";

Response: We thank the referee for this observation and we have made the change. Please see line 48.

  1. line 35 change "internal" with "splanchnic";

Response: The referee is right in suggesting this change and we have included it in the manuscript. Please see line 51.

  1. line 53 change "categories" with "hystotype";

Response: We thank the referee for this correction. Indeed it is more suitable and we have made the change in the text. Please see line 69.

  1. line 54 change "characteristic" with "features";

Response: Thank you for the suggestion. We have included it in the text. Please see line 69.

  1. lines 78-79 revising the sentence "forming?"

Response: We apologize for the mistake. We have made changes to this phrase to make it clear and correct. Please see line 97-98.

  1. line 91 change the sentence "A complete necropsy was performed, during which tissue sample were harvested" with the sentence "Necropsy was performed and tissue samples were collected";

Response: Thank you for the suggestion. We have made the changes accordingly. Please see lines 119-123.

  1. line 136-139 the sentence must be moved in chapter 2. Materials and Methods.

Response: We thank the referee for this correction. We have created a paragraph in the M&M section that includes the clinical examination. Please see lines 109-111, 119-123.

  1. line 137 change "manifesting" with "showing";

Response: We apologize for this mistake. We have corrected it in the text. Please see line 169.

  1. line 166 change "dissemination" with "pathway";

Response: We thank the referee for this suggestion. We have made the change. Please see lines 211, 228.

Round 2

Reviewer 3 Report

Dear Authors, the paper after major revision is really improved, but a minor revision is still needed after which the case report can be published. 

Please provide for the refinements of the text in the lines indicated.

Line 111 change the word after with later;

Line 119-123   from ... showing  to (figure 1d) ...must be moved to chapter Results;

Line 119 the sentence now reads - Necropsy was performed and tissue sample... ;

Line 128 change slide with Formalin Fixed Parffin Embedded (FFPE) sections;

Line 144 include pH value after (PBS); 

Line 154 change included with performed;

Line 178 change display with show;

Line 201 insert - neoplastic- between the ... disease;

Line 207 insert - of the right side - between node ... was;

Author Response

Response to Referee  #3

We thank referee #3 for his/her valuable suggestions that will certainly contribute to improving the quality of our paper.

Please find below the responses to the referee, point by point.

Line 111 change the word after with later;

Response: We thank the referee for this suggestion. We have made the changes in the text. Please see line 92.

Line 119-123   from ... showing  to (figure 1d) ...must be moved to chapter Results;

Response: We thank the referee for this observation. Indeed, this paragraph is more appropriate in the Results section. We have made the change in the text. Please see lines 100, 135-138.

Line 119 the sentence now reads - Necropsy was performed and tissue sample... ;

Response: The referee is right. After moving the above-mentioned paragraph to the Results section the phrase sounds much better. Please see line 100.

Line 128 change slide with Formalin Fixed Paraffin Embedded (FFPE) sections;

Response: We apologize for this omission. We have completed this information. Please see line 122.

Line 144 include pH value after (PBS);

Response: We apologize for this omission. We have added this information to the text. Please see line 119.

Line 154 change included with performed;

Response: We thank the referee for this observation. We have made the change in the text. Please see line 129.

Line 178 change display with show;

Response: We thank the referee for this suggestion. We have made the replacement in the text. Please see lines 21, 37, 150, 187, 229.

Line 201 insert - neoplastic- between the ... disease;

Response: We thank the referee for this suggestion. We have completed the sentence. Please see line 172.

Line 207 insert - of the right side - between node ... was;

Response: The referee is right. We have added this specification in the text. Please see line 178.
